# Diversity of “*Ca*. Micrarchaeota” in Two Distinct Types of Acidic Environments and Their Associations with *Thermoplasmatales*

**DOI:** 10.3390/genes10060461

**Published:** 2019-06-15

**Authors:** Olga V. Golyshina, Rafael Bargiela, Stepan V. Toshchakov, Nikolay A. Chernyh, Soshila Ramayah, Aleksei A. Korzhenkov, Ilya V. Kublanov, Peter N. Golyshin

**Affiliations:** 1School of Natural Sciences, Bangor University, Deiniol Rd, Bangor LL57 2UW, UK; f.bargiela@bangor.ac.uk (R.B.); s.ramayah@bangor.ac.uk (S.R.); p.golyshin@bangor.ac.uk (P.N.G.); 2Centre for Environmental Biotechnology, Bangor University, Deiniol Rd, Bangor LL57 2UW, UK; 3Winogradsky Institute of Microbiology, Research Center for Biotechnology, Russian Academy of Sciences, Prospect 60-Letiya Oktyabrya 7/2, Moscow 117312, Russia; stepan.toshchakov@gmail.com (S.V.T.); chernyh3@yandex.com (N.A.C.); kublanov.ilya@gmail.com (I.V.K.); 4National Research Center “Kurchatov Institute”, Akademika Kurchatova sq., 1, Moscow 123182, Russia; oscypek@yandex.ru

**Keywords:** “*Ca*. Mancarchaeum acidiphilum”, ARMAN-2, “*Ca*. Micrarchaeota”, DPANN superphylum, *Thermoplasmatales*, *Cuniculiplasma*, acidic environments, acid mine drainage sites, terrestrial hot springs

## Abstract

“*Candidatus* Micrarchaeota” are widely distributed in acidic environments; however, their cultivability and our understanding of their interactions with potential hosts are very limited. Their habitats were so far attributed with acidic sites, soils, peats, freshwater systems, and hypersaline mats. Using cultivation and culture-independent approaches (16S rRNA gene clonal libraries, high-throughput amplicon sequencing of V3-V4 region of 16S rRNA genes), we surveyed the occurrence of these archaea in geothermal areas on Kamchatka Peninsula and Kunashir Island and assessed their taxonomic diversity in relation with another type of low-pH environment, acid mine drainage stream (Wales, UK). We detected “*Ca*. Micrarchaeota” in thermophilic heterotrophic enrichment cultures of Kunashir and Kamchatka that appeared as two different phylotypes, namely “*Ca*. Mancarchaeum acidiphilum”-, and ARMAN-2-related, alongside their potential hosts, *Cuniculiplasma* spp. and other *Thermoplasmatales* archaea without defined taxonomic position. These clusters of “*Ca*. Micrarchaeota” together with three other groups were also present in mesophilic acid mine drainage community. Present work expands our knowledge on the diversity of “*Ca*. Micrarchaeota” in thermophilic and mesophilic acidic environments, suggests cultivability patterns of acidophilic archaea and establishes potential links between low-abundance species of thermophilic “*Ca*. Micrarchaeota” and certain *Thermoplasmatales*, such as *Cuniculiplasma* spp. *in situ*.

## 1. Introduction

Archaeal lineages represented in relatively low abundance in communities are still largely underexplored, which leaves many gaps in our knowledge regarding their distribution patterns, specific roles in the environment and interconnection with other community members. One example of these organisms, often referred to as microbial “dark matter”, is a group of Archaeal Richmond Mine Acidophilic Nanoorganisms (ARMAN), firstly detected in acid mine drainage systems of Iron Mountain (CA, USA) and later confirmed to be widely distributed in numerous low pH settings as well as in areas with neutral pH values [1,2,3,4,5,6,7,8]. Two names for candidate phyla were proposed for these lineages with significantly reduced genome sizes, namely “*Ca*. Micrarchaeota” and “*Ca*. Parvarchaeota” [1]. 

Based on the phylogenetic analysis of three concatenated protein sets, these groups are placed divergently [9]. “*Ca*. Micrarchaeota” were shown to cluster with “*Ca*. Diapherotrites” and “*Ca*. Parvarchaeota” to be associated with *Nanoarchaeota* [9]. Both groups are included into a tentative DPANN supercluster, incorporating also *Nanoarchaeota* and Candidate phyla “Aenigmarchaeota”, “Altiarchaeota”, “Diapherotrites”, “Huberarchaeota”, “Nanohaloarchaeota”, “Pacearchaeota”, and “Woesearchaeota” [10,11]. This group comprises organisms with reduced genome sizes and limited metabolic potential, small dimensions of cells, and reliance on symbiotic/mutualistic/commensalism-based interactions with other species for some of these archaea [10]. However, cultivated DPANN members were only recorded for *Nanoarchaeota* and “*Ca*. Micrarchaeota”. Cultured thermophilic associations between *Nanoarchaeota* and *Crenarchaeota* described and studied so far, are noteworthy to mention. The very first example of a co-culture was *Nanoarchaeum equitans* and *Ignicoccus hospitalis* [12], followed by an association between *Nanoobsidianus stetteri* and *Sulfolobales* Acd1 [13] a “*Candidatus* Nanopusillus acidilobi” and *Acidilobus* sp. 7a [14], and finally by a stable co-culture of “*Candidatus* Nanoclepta minutus” and *Zestosphaera tikiterensis* (representing a new member of the family *Desulfurococacceae*) [15]. For mesophilic and extremely acidophilic “*Ca*. Micrarchaeota” two cultured associations were documented: “*Ca*. Mancarchaeum acidiphilum” Mia14 with *Cuniculuplasma divulgatum* PM4 [6] and “*Ca*. Micrarchaeota”, ARMAN-1-related organism in a co-culture with consortium of two archaea (*Cuniculiplasma divulgatum*-like and another archaeon of the order *Thermoplasmatales*) together with a fungus [16]. Nevertheless, the distribution of “*Ca*. Micrarchaeota” in metagenomic data occurs at a global scale [8].

Estimation of phylogenetic diversity for “*Ca*. Micrarchaeota” [8] based on 16S rRNA gene sequence identity cut-offs for genus (94.5%) and family (86.5%) proposed by Yarza et al. [17] suggested clustering into 12 genera and 2 families. Relative abundance of “*Ca*. Micrarchaeota” in environmental microbiomes was shown to constitute <1% in 17 of 25 genomes reconstructed from metagenomic data, although an abundance of one assembly variant represented by FK_Sed_bin12_4 (Fankou AMD sediment, China) was as high as 5.3%–21.3% in certain subsamples [8]. Two acidophilic biofilm communities of Iron Mountain (CA, USA) accounted for ~7% “*Ca*. Micrarchaeota”/ARMAN-2 reads [2]. It was reported [8] that “*Ca*. Micrarchaeota” in comparison to “*Ca*. Parvarchaeota” possess wider distribution patterns since signatures of this group were detected not in acidic settings only, but in soils, peats, freshwater systems, and hypersaline mats. It may imply that “*Ca*. Micrarchaeota” demonstrates enhanced resistance to selective pressure, which enables their spreading and genome stability. Altogether, connection with *Thermoplasmatales* was predicted for 16 “*Ca*. Micrarchaeota” and for 13 “*Ca*. Parvarchaeota” species [8].

Analysis of 39 ARMAN-related genomes with different levels of completeness from various emplacements pointed at a certain variability in genome sizes (0.64–1.08 Mb) and predicted their potential involvement in carbon, nitrogen and iron cycles [8]. Furthermore, genomic analysis suggested microaerophilic and anaerobic lifestyles for “*Ca*. Micrarchaeota”, based on the aerobic respiratory chain, fermentation potential and presence of cytochrome *bd*-II encoding genes [8]. However, experimental confirmations of certain predicted physiological traits are still due [6,8,16]. 

To understand further patterns of distribution and phylogenetic variance of these archaea in particular environments, we focused our study on two types of acidic environments. 

We surveyed the presence of “*Ca*. Micrarchaeota” -related sequences in acidic geothermal areas via establishment of enrichment cultures with samples from terrestrial hydrothermal vents of Kamchatka Peninsula and Kunashir Islands (Russian Far East) and their consequent molecular study. We also address here the issue of relations between “*Ca*. Micrarchaeota” and *Thermoplasmatales* archaea, abundant in these niches and recovered in a number of enrichment cultures experiments. 

Another site used in this work was an extremely acidic low/moderate temperature environment: a shallow stream located in mine-impacted area of Parys Mountain (Parys Mt, Wales, UK), which was in focus of extensive studies for several years [6,18,19,20]. Metagenomic shotgun sequencing of Parys Mt acidic stream community revealed that among archaea *Euryarchaeota* accounted for 64%, “*Ca*. Micrarchaeota” (DPANN group) for 3% and “*Ca*. Parvarchaeota” for 0.2%, with the rest of the archaea making up 0.5 % of total prokaryotic reads [20]. Furthermore, the metagenomic study of microbiomes of sediment and water fractions from this site revealed the spatial distribution of “*Ca*. Micrarchaeota” to represent 0.3%–0.4% and 1.7%–1.4% of total reads in these microenvironments, respectively [20]. Here, we report on the assessment of diversity within “*Ca*. Micrarchaeota” at a local scale of Parys Mt acid mine drainage site. Diversity, heterogeneity, and similarity among “*Ca*. Micrarchaeota” variants in these two particular low-pH places differ in nature and genesis of acidity and will be discussed here.

## 2. Materials and Methods

### 2.1. Samples Collection and DNA Extraction

Samples of water/sediment from geothermal areas of Kunashir Island (Russia) were taken in July 2014 and were described to have further characteristics. KY1: A thermal vent in a solfataric area, with temporary coverage by lake water with parameters in a sampling spot of pH 2.1 and 58 °C (site O3, 43°52′38.67″ N 145°30′46.66″ E). KY2: a stream in a Northwest part of a fumarolic field, the sampling spot was located 3–5 m below fumaroles after junction of high-temperature acidic stream and cold neutral stream. The characteristics of the site were pH 2.3, 54 °C (43°59′15.72″ N 145°43′35.4″ E, 385 m). Additionally, KM3 enrichment was established from Kamchatka Peninsula geothermal site (Russia), characterised by pH 2.5 and 44 °C (54°30′4.61″ N 160°00′0.17″ E). These samples were used for the establishment of enrichment cultures and their consequent study via 16S rRNA gene clonal libraries and metabarcoding experiments. 

Samples of the top 1–3 cm water-saturated sediments of acidic stream and water and sediment as separate fractions located in Parys Mt (North Wales, UK, 53°23′13.6″ N 4°20′58.6″ W) have been taken in October of 2014 and in October 2016, respectively. Characteristics of the site were reported previously [20]. These samples (marked as Parys Mt clones) were used for bulk native DNA extraction, used for 16S rRNA gene amplicon clonal libraries and metabarcoding experiments. 

For enrichment cultures, the modified Medium 88 (DSMZ) was used [18]. Mixtures of beef extract and tryptone, each compound in concentration 1 g·L^−1^ were added to the medium. Additionally, the Medium 9K with supplements was used, as it was discussed previously [21]. The temperatures of incubation were 37, 45 and 50 °C. 

The DNA from all samples was isolated using MoBio Soil DNA isolation kit (QIAGEN). Selective targeting of archaea from enrichment cultures was done with specific primers. The primers used for archaeal amplification were archaeal Forward A23 (TCYGGTTGATCCTGCC) and Reverse universal R1492 (TACGGYTACCTTGTTACGACTT). For ARMAN and “*Ca*. Micrarchaeota” SSU rRNA gene amplification, designed pairs of primers MIAF (GCTTGGCGAATAAGTGCTGGGC) and MIAR (ATCTTGCGACCGTACTCCCCAG); and also ARM-MIA Fwd3 (GCGTACGGCTCAGTAACACGTAG) and ARM-MIA Rev (TTGAGGTGATCTATCCGCAGG) were used. The annealing temperature used in the PCR program with ARM/MIA primers was 60 °C. Generation of SSU rRNA gene clone libraries was done with TOPO TA 2.1 Cloning Kit for sequencing (Invitrogen, Carlsbad, CA, USA), according to the protocol of the supplier.

The procedures for preparation of V3-V4 rRNA gene amplicon libraries as well as the metabarcoding analysis pipeline were described previously [20]. Processing of Kunashir/Kamchatka samples was conducted in a similar way.

### 2.2. Phylogenetic Analysis

First, sequences of clones were matched against the NCBInr database using *blastn* algorithm [22]. Reference sequences for phylogenetic analysis were selected among the best hits based on their identity percentage. Final trees for *Thermoplasmatales* and “*Ca*. Micrarchaeota” were developed using a total number of 62 and 87 sequences, respectively. 

Both sets of sequences were aligned using *Mafft* v. 7 [23] with default parameters. Resulting multiple alignments were manually reviewed with *UGENE* [24] and finally trimmed using *Trimal* [25], removing those columns with more than 20% of gaps (gap penalty: 0.8) and similarity scores lower than 0.001.

Final phylogenetic trees were obtained using GTR as an evolution model by maximum likelihood method. Bootstrap was calculated using 1000 replicates. Generation of the final phylogenetic trees and final visualization have been both performed using scripts under *R* environment using *ape* [26] and *phangorn* [27] packages.

## 3. Results and Discussion

### 3.1. Kunashir and Kamchatka Samples

The optimal temperature for growth of all established low-pH (pH 1.2–2) enrichment cultures with samples from geothermal terrestrial hot springs from Kunashir and Kamchatka was revealed to be 50 °C. PCR amplification with *Archaea*-specific oligonucleotides revealed the presence of various *Thermoplasmatales* archaea and “*Ca*. Micrarchaeota” in heterotrophic enrichment cultures. A number of 16S rRNA gene amplicon libraries were established during the first year of cultivation to monitor the content of archaeal counterparts of three enrichment cultures (Figure 1, Figure 2 and Figure 3). The study of a variety of *Thermoplasmatales* archaea was needed for the understanding of interactions patterns with “*Ca*. Micrarchaeota” and perception of the phylogenetically structured ecological network within these settings. The bacterial components of cultures were not surveyed, and the general presence of bacteria was found not to be detectable after several transfers.

All enrichment cultures established in iron (II)-containing 9K medium (pH 1.7) showed the presence of *Acidiplasma* spp. [21] only; no other archaea of the order *Thermoplasmatales* or “*Ca*. Micrarchaeota” were detected. In relation to this, both iron oxidation capability in these particular “*Ca*. Micrarchaeota”-related organisms and their interactions with *Acidiplasma* spp. are very unlikely. Additionally, the reason for the lack of “*Ca*. Micrarchaeota” in iron-containing enrichment cultures might be that these certain conditions were favourable for *Acidiplasma aeolicum*-like archaea, but not other *Thermoplasmata*, on which “*Ca*. Micrarchaeota” might be metabolically dependent.

After six months of cultivation, heterotrophic enrichments (with amendments of beef extract and tryptone), showed the presence of *Thermoplasmatales*, namely *Acidiplasma* spp. and organisms distantly related to *Thermogymnomonas* and *Cuniculiplasma* (91%–93% 16S rRNA gene sequence identity). From all established heterotrophic variants, KY1, KY2 and KM3 enrichments showed the presence of “*Ca*. Micrarchaeota”, related to ARMAN-2 and Mia14-like containing clusters. Classifying the clones into OPUs (Operational Taxonomic Units), based on clones with ~99% of identity percentage among them, and checking the frequency of each OPU on each sample, we have noticed that OPUs 5 (*Cuniculiplasma*-related) and 8 (ARMAN-2 related) are clearly pronounced in KM3 and KY2 communities, respectively, whereas OPUs 1 (*Cuniculiplasma*-related) and 6 (*Acidiplasma*-related) are more present in the KY1 variant (Figure 4, Appendix A).

We have observed that KY1 and KM3 enrichment cultures were similar one with another and with Mia14-like archaea (100% 16S rRNA gene sequence identity). Organisms taxonomically similar to Mia14 and belonging most likely to the same genus, were also detected in metagenomic data in geothermal areas in China (TC_Endo_bin_32, Tengchong) and in different acid mine drainage systems [8]. According to phylogenetic analysis [8], this group represents the genus 9 within the Family 2 of “*Ca*. Micrarchaeota”. 

The clustering of KY2 and one of the sequences from KM3 with ARMAN-2 affiliated to genus 12/Family 2 according to [8] classifications was observed. This group includes a set of diverse sequences, also including two sets of metagenomic data TC_Endo_bin_6 and Me_Mat_bin1 from geothermal areas of Tengchong, China and Los Azufres National Park, Mexico, respectively. Other lineages from this cluster found in different AMD settings, including a significant proportion of PM clones, discussed further. 

We observed certain co-occurrence patterns between *Thermoplasmatales* archaea and “*Ca*. Micrarchaeota” in enrichment cultures. The KY1 enrichment culture was composed by *Acidiplasma* spp. and *Cuniculiplasma* spp. (97% sequence identity to *C. divulgatum*) archaea and Mia14-like “*Ca*. Micrarchaeota”. Another Kunashir Island enrichment KY2 showed the presence of three clades *Thermoplasmatales*. One variant showed sequence identity to *Acidiplasma aeolicum* of 99%, others were only distantly related to *Thermogymnomonas acidicola* and *Cuniculiplasma divulgatum* (89%–92%), and finally, the third group was represented by organisms with 97% SSU rRNA gene sequence identity to *C. divulgatum*. “*Ca*. Mancarchaeum acidiphilim” and ARMAN-2-related clades of “*Ca*. Micrarchaeota” were found in KY2 enrichment. The assessment of the content of Kamchatka enrichment culture (KM3) showed the presence of *Cuniculiplasma* spp. (with identity to *C*. *divulgatum* 97%), Mia14- and ARMAN-2-related clades of “*Ca*. Micrarchaeota” (Figure 1, Figure 2 and Figure 3). 

The content of enrichment cultures determined the presence of “*Ca*. Micrarchaeota” after 2 years of cultivation in minor numbers (0.3% reads), assessed by the DNA barcoding technique in a KY2 variant only. Initially, it was the most diverse enrichment culture, exemplified by three different *Thermoplasmatales* species. Among archaea in this particular variant, we identified sequences related to *Acidiplasma aeolicum*, 100% identity (35.8% reads) and sequences only distantly related to *Acidiplasma*
*aeolicum* (81% identity), sequences of the last lineage were not detectable earlier by clonal libraries approach. Other *Thermoplasmatales* archaea constituted 63.8% of all reads with *Cuniculiplasma* spp. in the variant KY2.2 A. Another parallel enrichment culture a KY2.2 B showed the presence of 4% reads of *Acidiplasma aeolicum* (95–100% identity) and other *Thermoplasmatales* in a number of 96.1% reads with 92% sequence identity with *Cuniculiplasma divulgatum*. 

Furthermore, after almost 3 years of cultivation, only two organisms persisted in a KY2.2.A: 87% of all reads were affiliated with *Acidiplasma aeolicum* (100% sequence identity), and 12% of reads to *Cuniculiplasma divulgatum* (97%). In KY2.2.B 94% reads were related to *Acidiplasma aeolicum* and 5% to *Cuniculiplasma divulgatum* with the same levels of identities of 16S rRNA gene sequence. The reason for the disappearance of “*Ca*. Micrarchaeota” from enrichment cultures after 3 years is not entirely clear since the possible host, *Cuniculiplasma* sp., which is able to provide essential metabolic precursors or other biomolecules was still present in the culture although in relatively smaller numbers being significantly outcompeted by *Acidiplasma* spp. It could be that for some reasons, higher densities of host cells are needed to support the life of “*Ca*. Micrarchaeota”. We cannot also exclude the possible involvement of *Acidiplasma* species into the interaction with “*Ca*. Micrarchaeota”. However, this needs further experimental confirmation.

The detection of “*Ca*. Micrarchaeota” in samples from geothermal sites of Kunashir and Kamchatka was intriguing. The occurrence of *Thermoplasmatales* was rather expected, since these organisms are known inhabitants of Kamchatka hot springs and were detected to make up to 39% of all archaea in groundwater microbiome (pH 4.0, 50 °C) in the East Thermal Field of Uzon Caldera [28]. However, the predominance of *Acidiplasma* representatives, which probably cannot interact with “*Ca*. Micrarchaeota” poses the question of who the “*Ca*. Micrarchaeota” actual host is, since the temperatures of incubation of Kamchatka and Kunashir enrichments (45 and 50 °C) seem to be too high for *Cuniculiplasma divulgatum* representatives.

*Thermoplasmatales*, namely *Thermoplasmatales* group A10, were previously found in quite significant numbers (up to 52% of total communities) in Kamchatka hot springs Kaskadny and Arkashin Shurf characterized by moderate acidity and temperatures [29]. Moreover, the presence of *Nanoarchaeota* in Kamchatka hot springs was also confirmed, yet no information about “*Ca*. Micrarchaeota” was presented [29]. These results emphasize the demand in further community studies of these acidic ecosystems to understand abundance, diversity and specific correlations with environmental variables for minor groups, such as “*Ca*. Micrarchaeota”, in microbiomes.

### 3.2. Diversity of “Ca. Micrarchaeota” in Parys Mt AMD

Our results of the analysis of Parys Mt AMD 16S rRNA gene sequences have revealed significant diversity of “*Ca*. Micrarchaeota”-related groups in Parys Mt environment. Altogether, we detected a number of variants with a variety of affiliations to families and genera proposed by phylogenetic analysis [8].

The first cluster represented by numerous PM clones showed relatively low sequence identity to other “*Ca*. Micrarchaeota”-like variants and belongs to genera 3–4 (Family 1), according to the classification of [8]. The nearest sequence to this PM cluster was represented by the Micrarchaeota FK_AMD_bin113 from Fankou AMD outflow (China) (Figure 1). We have also identified the most similar sequence to this group with an ID GQ141775 from filamentous mat from an acidic stream of the Rincon de la Vieja Volcano National Park (Costa Rica), which exhibited 99% sequence identity with 50% coverage. 

The single PM clone 18 with sequence ID MH463124 was phylogenetically located outside this cluster (Figure 1) and was similar to the FK_AMD_2010_bin_24, from Fankou mine tailings AMD outflow, referred to as the Genus 2 within the Family 1 [8]. 

The Family 2 proposed by [8] includes two clusters and three singletons of PM “*Ca*. Micrarchaeota”-related sequences. One clone 23 (MH463107) was shown to be distantly related to other “*Ca*. Micrarchaeota” (Figure 1), another clone 1 (MH463092) clusters together with “*Ca*. Mancarchaeum acidiphilum” Mia14-related cluster and is associated with those from enrichment KY1 and KM3 enrichment cultures. Finally, the third separately placed PM clone (MH463104) was derived from ARMAN-2–related organisms, Kunashir (KY1) and Kamchatka (KM3) variants and many other sequences from Fankou AMD sediment, Tengchong geothermal area (both, China) and in genomic data from acidic stream of the Rincon de la Vieja Volcano National Park (Costa Rica). Additionally, similar sequences were identified in metagenomic data from acidic biofilm (KC127696) Harz Mountains, Germany [3] and sequences (JF280280/29) from microbial community of hot spring of the Colombian Ands [30]. Another large cluster of PM clones was affiliated to ARMAN-2 group (Figure 1).

Interestingly, in Iron Mountain AMD site (CA, USA) “*Ca*. Micrarchaeota” was represented by three groups, categorised as ARMAN-1-3, together with “*Ca*. Parvarchaeota” lineages, with ARMAN-2 group being the most abundant in seven biofilms studied [2]. Another surveyed AMD site Los Rueldos (Spain) showed the diversity of “*Ca*. Micraerchaeota” represented by two clusters related to ARMAN-1 and 2 lineages [4]. These data are in the agreement with our conclusions on the presence of ARMAN-2-like organisms in both studied sites.

These versatile clusters of “*Ca*. Micrarchaeota” in Parys Mt environment may rely on interactions with different archaea. Firstly, the dependency of “*Ca*. Micrarchaeota” on symbiotic interactions with *Cuniculiplasma* and *Cuniculiplasmataceae*, as previously reported [6], could be confirmed. Bearing in mind the abundance of archaea of the order *Thermoplasmatales* in this ecological niche (58% of all reads as unclassified *Thermoplasmatales* and for 4% as *Cuniculiplasmataceae* [20]) one could suggest further potential hosts from this group to support the needs of diverse members of “*Ca*. Micrarchaeota”. 

The dependency of ARMAN archaea on multiple hosts was proposed from analysis of 39 “*Ca*. Micrarchaeota” genomes that lack the genes for amino acids and nucleotides synthesis [8]. The proposition on multiple hosts was also outlined for *Nanoarchaeota* [31]. Our previous data on the composition of enrichment cultures established with the sediment of Parys Mt also suggested that one species of *Cuniculiplasma* spp. could be a host for different ARMAN organisms (Mia14- and ARMAN-1-related species) [20]. “*Ca*. Mancarchaeum acidiphilum Mia14” metagenomic reads together with ARMAN-1-related “A-DKE” sequences were detected, favouring heterotrophic, aerobic and mesophilic cultivation conditions [20]. Additionally, the earlier study [16] reported on the stable culture consisting of “*Ca*. Micrarchaeota” spp., two *Thermoplasmatales* archaea, one of which was *Cuniculiplasma divulgatum*, and a fungus. CARD-FISH experiments suggested localisation of ARMAN cells with both *Thermoplasmatales* organisms [16]. To sum up, the above data advocate for a range of conditions supporting the life of diverse “*Ca*. Micrarchaeota” in the ecosystem. 

## 4. Conclusions

We revealed the presence of “*Ca*. Micrarchaeota” in geothermal areas of Kunashir and Kamchatka. The content of moderately thermophilic heterotrophic enrichment cultures suggests *Cuniculiplasma* spp., *Acidiplasma* spp. and other *Thermoplasmatales* without defined taxonomic positions being potential hosts for “*Ca*. Micrarchaeota”. The assessment of the diversity of two types of acidic sites, thermal springs, and the AMD stream revealed significant variety among “*Ca*. Micrarchaeota”. Two phylotypes found to be widely represented in geothermal areas, namely “*Ca*. Mancarchaeum acidiphilum” and ARMAN-2-related, were detected also in AMD ecosystems, confirming ubiquity and abundance for these archaeal groups. Five different clusters of sequences were observed for Parys Mt “*Ca*. Micrarchaeota” lineages with 88% 16S rRNA gene sequence identities between the most distant groups. The present study is a step forward towards understanding of diversity and cultivability of low-abundance microbial species in thermal and low/moderate temperature acidic environments. We also think that these results expand our knowledge on the cultivability patterns of acidophilic archaea of the order *Thermoplasmatales* from geothermal acidic placements and emphasise the remarkable co-occurrence of “*Ca*. Micrarchaeota” and certain *Thermoplasmatales* species, which points at their interactions in situ.

## Figures and Tables

**Figure 1 genes-10-00461-f001:**
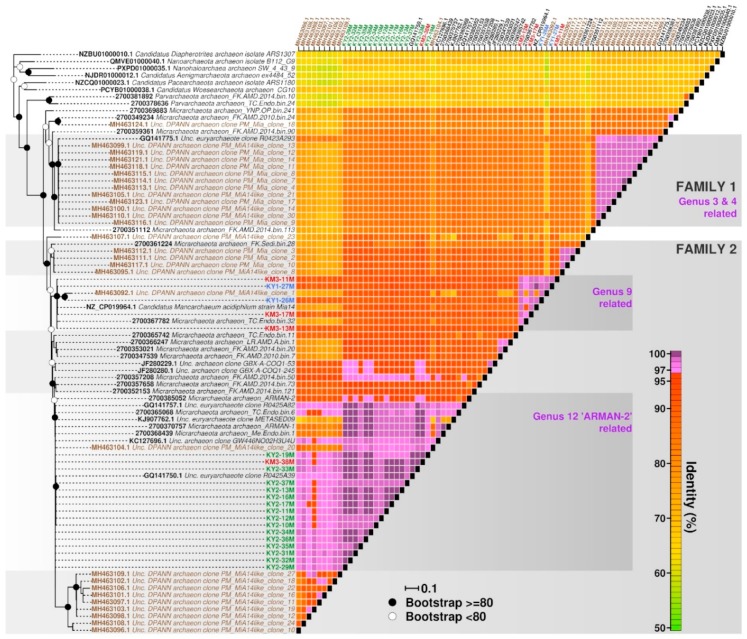
Diversity of “*Ca*. Micrarchaeota”-related clones, their phylogenetic affiliation, and identity level in the enrichment cultures established with Kunashir and Kamchatka samples, and from Parys Mt metagenome. Kunashir clones (KY1 and KY2) are in green and blue, respectively; Kamchatka clones (KM) are in red, Parys Mt clones are in brown.

**Figure 2 genes-10-00461-f002:**
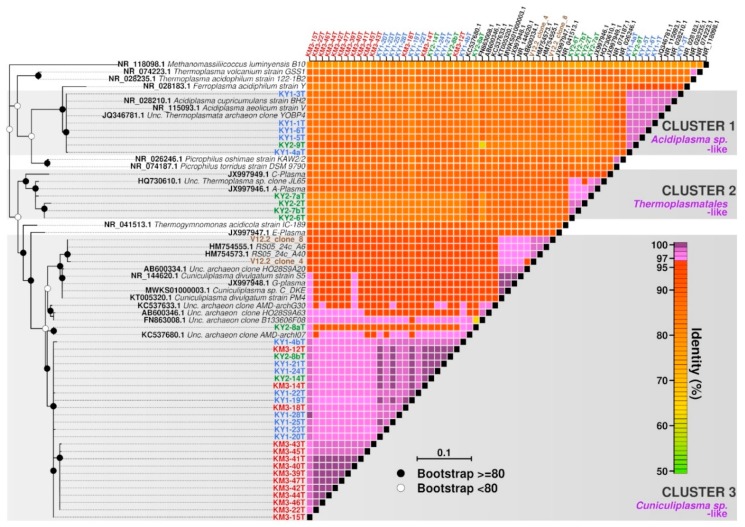
Diversity of *Thermoplasmatales*-related clones, their phylogenetic affiliation, and sequence identity level in enrichment cultures established with Kunashir (KY1 and KY2) and Kamchatka (KM3) samples. Kunashir clones (KY) are in green and blue, respectively; Kamchatka clones (KM) are in red.

**Figure 3 genes-10-00461-f003:**
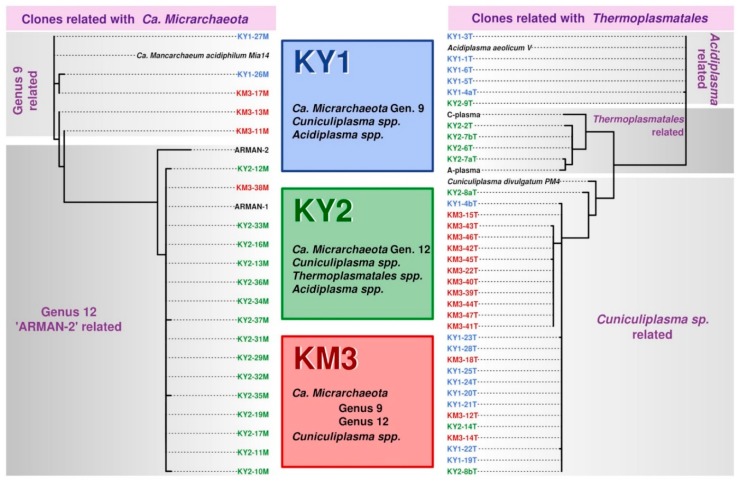
Representation of archaea (“*Ca*. Micrarchaeota” and *Thermoplasmatales*) in enrichment cultures established with Kunashir and Kamchatka samples.

**Figure 4 genes-10-00461-f004:**
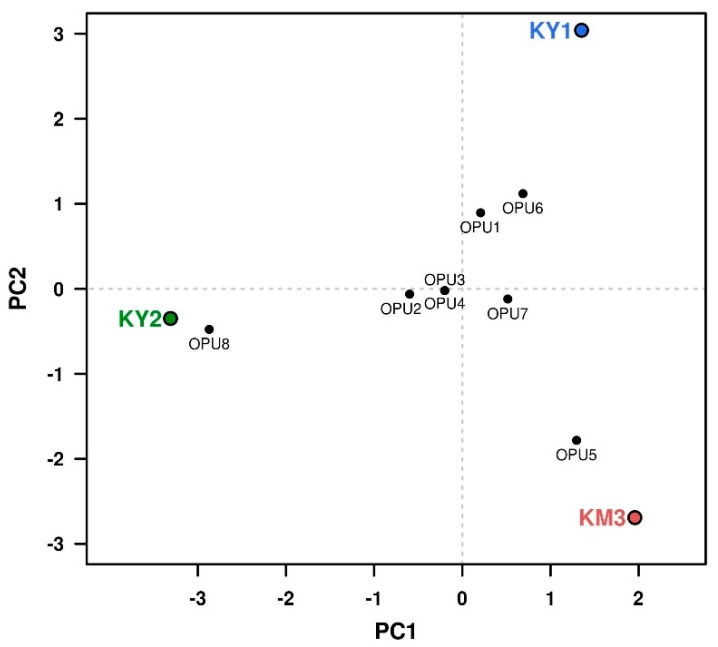
Principal component analysis (PCA) of the community profile for the clone libraries from the three different areas, where the first (PC1) and second (PC2) components would explain the 67.04% and 32.96% of the variability, respectively. Operational Phylogenetic Units (OPUs) have been assigned to groups of clones with sequence identity ~99% or higher. The figure has been developed under an R programming environment.

## Data Availability

The sequences were submitted to NCBI GenBank under accession numbers MK578767-MK578823.

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
