# Peer review of "Diversity of “*Ca*. Micrarchaeota” in Two Distinct Types of Acidic Environments and Their Associations with *Thermoplasmatales"

_genes, 2019, doi:10.3390/genes10060461_

Round 1
Reviewer 1 Report
The manuscript by Golyshina et al describes the diversity of “Micrarachaeota” a recently proposed group of Archaea that remains poorly understood. The authors analyzed samples from locations in Russia and the UK and performed laboratory enrichments. Phylogenetic relationships between the various lineages were determined and the relative abundance of the various target species studied and compared with those of other archaea in the cultures. In the end, however, the host(s) of the different Micrarchaeota were not confidently identified and some lineages were lost.
Overall the manuscript is well written and present novel and useful information about the biodiversity of this lineage of Archaea as well as steps towards cultivation together with some potentially novel hosts. I am disappointed though that no images of the cultures were presented. After several years of maintaining enrichments, don’t the authors have some FISH images, or perhaps some electron micrographs to show? Why no microbiology data?
Also, while the authors characterized samples from Russia and the UK, only the results from Russia are presented in the Abstract. That should be rectified to summarize the entire study.
Line 38 . “minor quantities” is not a proper term. I suggest replacing it by “relatively low abundance”
Line 41. Microbial “dark matter”
Line 44. Those candidate phyla name are not official and recognized and therefore you cannot say “granted”, replace with “proposed”
Lines 60-66, correct italicized text.
Line 123 shotgun
Line 152 obtention?
Author Response
The manuscript by Golyshina et al describes the diversity of “Micrarachaeota” a recently proposed group of Archaea that remains poorly understood. The authors analyzed samples from locations in Russia and the UK and performed laboratory enrichments. Phylogenetic relationships between the various lineages were determined and the relative abundance of the various target species studied and compared with those of other archaea in the cultures. In the end, however, the host(s) of the different Micrarchaeota were not confidently identified and some lineages were lost.
Overall the manuscript is well written and present novel and useful information about the biodiversity of this lineage of Archaea as well as steps towards cultivation together with some potentially novel hosts. I am disappointed though that no images of the cultures were presented. After several years of maintaining enrichments, don’t the authors have some FISH images, or perhaps some electron micrographs to show? Why no microbiology data?
We completely agree with these comments and the importance of EM or FISH images for understanding interactions between Thermoplasmatales and “Ca. Micrarchaeota”. In this relation, I would like to point that cultivation of these organisms in the lab is extremely difficult and both techniques require significant quantities of biomass. To unambiguously confirm the identity of cells of “Ca. Micrarchaeota” antibody-based detection and visualization techniques would be necessary, which needs identification of target proteins via genome sequencing. In the course of this work, cultures were monitored continuously via phase-contrast microscopy, however this does not provide sufficient information about morphology and occurrence of different organisms in cultures keeping in mind a broad range of cell sizes of archaea of the order Thermoplasmatales and no clarity about cell structures/sizes of “Ca. Micrarchaeota”.
Also, while the authors characterized samples from Russia and the UK, only the results from Russia are presented in the Abstract. That should be rectified to summarize the entire study.
Done
Line 38 . “minor quantities” is not a proper term. I suggest replacing it by “relatively low abundance”
Done
Line 41. Microbial “dark matter”
Done
Line 44. Those candidate phyla name are not official and recognized and therefore you cannot say “granted”, replace with “proposed”
Done
Lines 60-66, correct italicized text.
We consider that the writing style of organisms with Candidatus status must be followed as recommended by Euzeby taxonomy resource http://www.bacterio.net/-foreword.html#candidatus
“The names included in the category Candidatus are usually written as follows: Candidatus (in italics), the subsequent name(s) in roman type (with an initial cap for the first subsequent name or the single subsequent name) and the entire name in quotation marks. For example, "Candidatus Phytoplasma", "Candidatus Phytoplasma allocasuarinae"….”
Line 123 shotgun
It was deleted now.
Line 152 obtention?
The word was changed to generation.
Reviewer 2 Report
This manuscript describes the diversity of nanoarchaeotal clades in acidic environments and their phylogenetic relationship to known taxa, according to 16S rRNA gene sequences. The authors have do e a huge job with cultivating and enriching the archaea and found evidence on the nanoarchaeal lineages to live in close association with thermoplasmatales hosts. The text is quite clear, although it should be language checked. In addition, I have these specific concerns;
Check the use of 16S rRNA and 16S rRNA gene throughout the text. I’m sure you have cloned the gene, not the RNA.
What was the shotgun sequenceng for? My interest was immedietly hightened when I read that you had done shotgun sequencing, but I was really dissapointed to find that you had only looked at the 16S rRNA genes from these. Why? You draw conclusions about the preferences and gene content of the nanoarchaea based on literatuse, but not based on your sequence data? Was there a difference in the 16S rRNA genes you fot from the shotgun seq data compared to the clones and amplicon data? Either add the shotgun dat to this manuscript, or remove it.
In addition, I would like to see the community profiles you got from the clone libraries and the different primers for the barcoded amplicon sequencing as well as richness and diversity indices from these results. I would also like to see statistical evidence of how the communities differ between the different sites and the different detection methods.
L124 - are these all modifications?
Author Response
This manuscript describes the diversity of nanoarchaeotal clades in acidic environments and their phylogenetic relationship to known taxa, according to 16S rRNA gene sequences. The authors have do e a huge job with cultivating and enriching the archaea and found evidence on the nanoarchaeal lineages to live in close association with thermoplasmatales hosts. The text is quite clear, although it should be language checked. In addition, I have these specific concerns;
We are very much thankful for this positive assessment of our work.
Check the use of 16S rRNA and 16S rRNA gene throughout the text. I’m sure you have cloned the gene, not the RNA.
Done
What was the shotgun sequenceng for? My interest was immedietly hightened when I read that you had done shotgun sequencing, but I was really dissapointed to find that you had only looked at the 16S rRNA genes from these. Why? You draw conclusions about the preferences and gene content of the nanoarchaea based on literatuse, but not based on your sequence data? Was there a difference in the 16S rRNA genes you fot from the shotgun seq data compared to the clones and amplicon data? Either add the shotgun dat to this manuscript, or remove it.
We removed this part.
Shotgun sequencing of fragment metagenomic libraries, as well as sequencing of certain Parys Mountain enrichment culture was described previously by Korzhenkov et al., and used in this particular study only as a reference.
In addition, I would like to see the community profiles you got from the clone libraries and the different primers for the barcoded amplicon sequencing as well as richness and diversity indices from these results. I would also like to see statistical evidence of how the communities differ between the different sites and the different detection methods.
We completely agree with the importance of community profiles for the characterization of microbial communities in microbial ecology studies. Nevertheless, the main scope of this paper is qualitative description of the diversity of “Ca. Micrarchaeota” in two different and geographically isolated environments. According to this aim, the design of this study is not universal and targets only two distinctive taxonomic groups. Thus, primers, used for the amplification of “Ca. Micrarchaeota” were highly specific and therefore produce results, which might confuse the reader regarding the overall structure of the communities. Also, a low number of clones, as opposed to high numbers of metabarcoded reads, would not be statistically robust.
On the other hand, the full length 16S rRNA gene clones provide much better taxonomic resolution than short V3-V4 amplicons and therefore fits better for the aim of this paper.
However, Principal component analysis (PCA) of the community profile for the clone libraries from the three different areas, where the first (PC1) and second (PC2) components would explain the 67.04 % and 32.96% of the variability, respectively, was done and included now as the Fig. 4 and the Supplementary Table S1. Operational Phylogenetic Units (OPUs) have been assigned to groups of clones with sequence identity ~99% or higher. We therefore ask for the further advice from the Reviewer and Editor whether this Figure is demonstrative enough and might be kept here
L124 - are these all modifications?
Yes, all modifications were the same as it was described in the paper of Golyshina et al., 2016.
Round 2
Reviewer 2 Report
The manuscript has been updated and edited quite well. I'm happy to see the new Figure 4. I still need you to describe what the data points in the figure are in the figure legend, and what you mean with OPU. I did not find OPU explained anywhere in the text. You need to give some more information about what an OPU is, what it is constituted of, how the sequences are grouped into OPUs, and of course explain the abbreviation in the text, not only in the figure legend. You also need to include the description for the PCA analysis in the materials and methods.